# Phytoremediation of Heavy Metal Pollution: A Bibliometric and Scientometric Analysis from 1989 to 2018

**DOI:** 10.3390/ijerph16234755

**Published:** 2019-11-27

**Authors:** Chen Li, Xiaohui Ji, Xuegang Luo

**Affiliations:** 1School of Chemistry and Environmental Science, Shaanxi University of Technology, Hanzhong 723001, China; leechen_317@126.com (C.L.); jixh@snut.edu.cn (X.J.); 2School of Environment and Resource, Southwest University of Science and Technology, Mianyang 621010, China; 3Shaanxi Key Laboratory of Catalysis, Shaanxi University of Technology, Hanzhong 723001, China; 4School of Life Science and Engineering, Southwest University of Science and Technology, Mianyang 621010, China

**Keywords:** CiteSpace, research hotpots, research trends, phytoremediation, heavy metal

## Abstract

This paper aims to evaluate the knowledge landscape of the phytoremediation of heavy metals (HMs) by constructing a series of scientific maps and exploring the research hotspots and trends of this field. This study presents a review of 6873 documents published about phytoremediation of HMs in the international context from the Web of Science Core Collection (WoSCC) (1989–2018). Two different processing software applications were used, CiteSpace and Bibliometrix. This research field is characterized by high interdisciplinarity and a rapid increase in the subject categories of engineering applications. The basic supporting categories mainly included “Environmental Sciences & Ecology”, “Plant Sciences”, and “Agriculture”. In addition, there has been a trend in recent years to focus on categories such as “Engineering, Multidisciplinary”, “Engineering, Chemical”, and “Green & Sustainable Science & Technology”. “Soil”, “hyperaccumulator”, “enrichment mechanism/process”, and “enhance technology” were found to be the main research hotspots. “Wastewater”, “field crops”, “genetically engineered microbes/plants”, and “agromining” may be the main research trends. Bibliometric and scientometric analysis are useful methods to qualitatively and quantitatively measure research hotspots and trends in phytoremediation of HM, and can be widely used to help new researchers to review the available research in a certain research field.

## 1. Introduction

Rapid urbanization and industrial development and the intensification of agriculture not only support the local and national economy but also instigate many environmental pollution problems [1,2]. Heavy metal (HM) pollution mainly comes from the mining and smelting of metals, from electroplating, and from other industrial processes [3,4]. The excessive application of fertilizers and pesticides and irrigation with sewage in modern agriculture also leads to HM pollution in farmland and water [5,6]. HMs in the environment usually have the characteristics of being bioaccumulative and physiotoxic and can persist indefinitely; they can enter organisms through the food chain and result in serious negative impacts on the environment [7]. Cost-effective and efficient remediation technology for HM pollution has been receiving worldwide attention [8,9]. Phytoremediation, which is a cheap and sustainable pollution remediation technology, originated in the early part of the 20th century and was recognized as a competitive solution for HM contamination [10,11].

After nearly 70 years of development, the number of publications on the topic of phytoremediation of HMs has continued to increase at a stable pace. Phytoremediation has been widely applied to water, air, and soil pollution remediation [12,13,14]. Although phytoremediation of HM pollution has received considerable attention [15,16], the research hotspots and trends have rarely been studied systematically. Moreover, no previous studies have analyzed its research corpus to such depth to include aspects such as key-words or co-citation clusters. At present, the phytoremediation of HMs has reached a critical period for large-scale field application and commercialization. Thus, it is imperative to create a summary of the current status with emerging trends and vital turning points in the field of phytoremediation. As a consequence of the stage of phytoremediation research, we need these synthesis data [17,18,19]. Bibliometric analysis—a quantitative method combining mathematical and statistical analyses—can assess the developmental trends in a research field by analyzing the published scientific literature. This method can help researchers quickly grasp the evolution of the characteristics of a research theme over time, which not only greatly improves the efficiency of scientometric research, but can guide subsequent research. In this research, a scientometric visualization software—CiteSpace—was used as a text mining and visualization tool for bibliometric analysis [20]. The purpose of this study is to provide a comprehensive and systematic review of the application of phytoremediation to HM pollution with CiteSpace [21] and explore the research hotspots and trends of phytoremediation in HM pollution [20].

## 2. Materials and Methods

### 2.1. Data Collection and Processing

In view of the special requirements of CiteSpace for data structure and content, the data used for the analysis were collected from the following Web of Science Core Collection (WoSCC) indices: the “Science Citation Index-Expanded (SCI-E) (1900–2018)”, “Social Sciences Citation Index (SSCI) (1900–2018)”, “Conference Proceedings Citation Index-Science (CPCI-S) (1990–2018)”, “Conference Proceedings Citation Index-Social Science and Humanities (CPCI-SSH) (1990–2018)”, “Emerging Sources Citation Index (ESCI) (2015-2018)”, “Current Chemical Reactions (CCR-Expanded) (2003–2018)”, and “Index Chemicus (IC) (2003–2018)”. These databases are the most important and commonly used in the field of natural science. The searches were as follows: “TS = (“phytoextraction*” OR “phytostabilization*” OR “phytodegradation*” OR “phytostimulation*” OR “phytovolatilization*” OR “phizofiltration*” OR “phytodesalination*” OR “accumulator plant*” OR “Hyper*accumulator*”) AND TS = (“trace metal*” OR “heavy metal*” OR “heavymetal*” OR “trace element*” OR “cadmium” OR “Cd” OR “copper” OR “Cu” OR “lead” OR “Pb” OR “zinc” OR “Zn” OR “nickel” OR “Ni” OR “mercury” OR “Hg” OR “arsenic” OR “As” OR “Chromium” OR “Cr”)”. Finally, whole records, including titles, abstracts, and cited references, downloaded as plain text, formed the local database for subsequent analysis with CiteSpace (5.5.R2). The records were preprocessed with the deduplication function of CiteSpace.

### 2.2. Scientometrics Analysis Methods

In this research, a freely available, Java-based scientific visualization software package, CiteSpace, was utilized to visualize the networks [20]. Co-occurrence analysis and co-citation analysis were used to obtain a network map of the knowledge structure, and cluster analysis and burst detection were used to identify the research hotspots and trends. The status and development trend of the use of phytoremediation for HMs were displayed in a panorama. In a particular field of research, a citation burst indicates the appearance of an active topic. Similarly, in the results of a CiteSpace analysis, the appearance of a burst in a network indicates that a node has surged in a period of time. For example, a citation burst means that a publication has attracted extraordinary attention from the scientific community. Furthermore, a cluster that contains a series of nodes with sharp citation bursts indicates an emerging trend or an active area of research. Thus, in this research, clustering analysis and burst detection were used to identify abrupt changes in events and determine the emerging trend of the phytoremediation of HMs [22].

Burstness is a measure of the rate of abrupt change in citation frequency over time. A node with strong burstness is marked with a red center, indicating a sharp increase in the citation frequency in a particular time period [23]. Betweenness centrality (BC) is an indicator used for describing the importance of nodes in a network. A node with high BC (≥0.1) is defined as a turning point and marked with purple circles [23,24]. The thickness of the purple circle is proportional to the BC score. A turning point links the conceptual clusters found at different times and can help us determine the evolution of the knowledge domain over time. Equation (1) shows the method of calculating BC. In Equation (1), *g_st_* represents the number of shortest paths between node *s* and node *t*, and $n_{\mathrm{st}}^{i}$ is the number of those paths that pass through node *i*. Nodes with large BC scores are usually the connecting channels between nodes, and a large BC score normally indicates the important position of the node in the network [25].
(1)$$BC_{i}=\sum_{s\neq i\neq t}\frac{n_{st}^{i}}{g_{st}}.$$

Frequency is a counting index in the network analysis results, and it reflects the co-occurrence/co-citation frequency of the published articles in the local database [26]. In the network clustering results, the modularity (*Q*) and the mean silhouette (*S*) are two important parameters for measuring the quality of the clustering, revealing the “overall structural properties” of the clusters in the network, ranging from 0 to 1. In the cluster map, greater *Q* values indicate better clusters of nodes, and *Q* > 0.3 indicates that the community structure of the network is significant. Similarly, greater *S* values denote higher homogenization of the nodes in a cluster, and S > 0.7 generally suggests that the cluster has high credibility [26].

In a co-operation/co-citation analysis, the time slice was set as one year, and the top 50 articles from each time slice were selected for analysis. CiteSpace provides functions to reduce the number of links while retaining the most salient structure. The link reduction functions include a “minimum spanning tree pruning” and a “pathfinder network scaling pruning”. In this research, the links that have little influence on the most significant structure were reduced with “pathfinder network scaling pruning”. On the basis of the bibliographic records retrieved from the WoSCC, the publishing characteristics of the research on the topic of phytoremediation in HM pollution were studied by the scientometric analysis method. On the basis of the CiteSpace co-occurrence and co-operation analysis, the national academic cooperation characteristics and interdisciplinary characteristics of phytoremediation of HMs were obtained. Moreover, the hotspots and trends of phytoremediation of HMs were obtained with the CiteSpace co-citation and burst detection functions [27,28,29]. At the same time, Bibliometrix, open-source software for text mining that runs in Rstudio, was used based on the word cloud formed from the keywords, keywords plus, titles, and abstracts. The detailed research plan is shown in Figure 1.

## 3. Results

### 3.1. Characteristics of Publication Outputs

A total of 6873 records related to phytoremediation of HMs were identified from 1945 to 2018, and they were classified into 12 document types (article, proceedings paper, review, meeting abstract, etc.). The body of literature was mainly composed of articles (6116), followed by proceedings papers (570), and reviews (407). The inter-annual total publications (TP) of all types increased from 1 in 1945 to 695 in 2018, and the TP of articles increased from 1 to 593. The inter-annual variation in publications indicated that research on the phytoremediation of HMs was first published in 1945, but that consistent publication has been gradually formed since 1989. From 1989 to 2018, the number of TPs showed a significant ascending curve. The publication output performance from 1989 to 2018 is shown in Figure 2. The number of TPs in 2018 (695) was approximately 18 times that of 1999 (38). In particular, the TP from 2011 to 2018 accounted for 61.78% of all the studies on the phytoremediation of HMs. According to the characteristics of the publication outputs, a research period between 1989 and 2018 was selected and divided into three research periods (1989–1999, 2000–2010, and 2011–2018) (Figure 3). 

On the basis of the set retrieval formula, literature retrieval from the WoSCC database was carried out. After retrieving the literature, the published data were statistically analyzed by bibliometrics. After the data analysis, the trend of the number of publications changing with time was obtained. According to Figure 2, from 1989 to 2018, the number of publications non-linearly increased over time. Since 2000, publications related to the phytoremediation of HMs have shown a rapid growth trend, especially during 2011 and 2018, where the number of publications showed an extreme increase. This indicated the continuing concern of the academic community in this field. In each research period (Figure 3), the linear model can better describe the annual publications, which can describe the research trend of phytoremediation of HMs.

### 3.2. Subject Categories Co-Occurrence Analysis

An article that is indexed by the Web of Science(WoS) usually belongs to one or more subject categories. The co-occurrence analysis of subject categories contributes to detecting the disciplines involved in intellectual development and the interdisciplinary characteristics of a specific knowledge domain [30]. Phytoremediation technology has obvious interdisciplinary characteristics; for example, it is well known to the public and involves plant science, environmental science, and agricultural science. In this section, the co-occurrence analysis of the subject category was used to obtain detailed categories involved in the research of phytoremediation of HMs and determine its interdisciplinary characteristics. In this research, the node/link colors were used to signify the corresponding year from 1989–2018, and the gray color from light to dark corresponds with 1989–2008, whereas purple to red corresponds with 2009–2018 (Figure 4). It is worth noting that the links’/nodes’ colors in Figures 5 and 7–9 have the same means. 

A total of 6842 records in the 6881 search results had valid subject categories. These subject categories belonged to 117 unique subject categories. A total of 68 nodes and 93 links were identified in the subject categories co-occurrence network. As shown in Figure 5, we found that research on phytoremediation of HMs is multifaceted and covers a wide range of interests. Environmental Science and Plant Sciences are two primary subject category groups and have the characteristics of being multidisciplinary. The environmental science category group mainly includes “Environmental Sciences & Ecology”, “Environmental Sciences”, “Engineering, Environmental”, and “Toxicology”, whereas the plant sciences category group mainly includes “Soil Science”, “Plant Sciences”, “Agriculture”, and “Agronomy”. An increasing number of subject categories have been involved in this research field in recent years [31,32,33]. According to the co-occurrence analysis of the subject category, “Environmental Sciences & Ecology”, “Environmental Sciences”, and “Plant Sciences” are obviously larger than other nodes, have more frequent co-occurrences, and can be identified as research hotspots among the subject categories.

On the basis of the colorful circle analysis, the earliest research was mainly conducted in the “Plant Sciences”, “Environmental Sciences & Ecology”, and “Environmental Science” fields, which had gray inner circles. On the other hand, nodes such as “Water Resources”, “Mining & Mineral Processing”, and “Mineralogy” had colorful inner circles, indicating that they were recently incorporated into the research of phytoremediation of HMs.

Furthermore, we also found that nodes such as “Chemistry, Inorganic & Nuclear”, “Chemistry”, “Biochemistry & Molecular Biology”, “Ecology”, and “Chemistry, Multidisciplinary” were marked with thick purple outer circles; they are connected to multiple nodes or serve as channels for other nodes, indicating that they play an important role in the subject category co-occurrence network. Without these nodes, the whole network would become very loose, with isolated nodes.

The top 10 most frequent categories and the turning points in the network according to the subject category co-occurrence analysis result are listed in Table 1 and Table 2. Furthermore, among the 117 unique subject categories, occurrence bursts were detected. Bursts were found in 25 subject categories, and the nodes with strong bursts are listed in Table 3.

While the co-occurrence frequency of nodes represents the output efficiency of papers in a given category, the BC score means that the papers published in that category are given attention by the scientific peers involved in the topic of the category. Therefore, categories that have both high-frequency and high BC scores are category hotspots in the phytoremediation of HMs. Thus, according to Table 1 and Table 2, categories such as “Environmental Sciences & Ecology”, “Plant Sciences”, and “Agriculture” are basic supporting categories of research on phytoremediation of HMs. Categories such as “Chemistry, Inorganic & Nuclear”, “Biochemistry & Molecular Biology”, “Chemistry”, “Ecology”, “Chemistry, Multidisciplinary”, “Mining & Mineral Processing”, “Engineering, Chemical”, “Metallurgy & Metallurgical”, and “Engineering” also play a vital role in the interdisciplinary research of phytoremediation of HMs.

According to the burst detection results in Table 3, “Plant Sciences” had the largest burst strength at 57.7754, the earliest burst beginning year (1991), and the longest burst duration of 16 years. This was followed by “Agronomy”, which had a burst beginning at the year of 1997, a burst strength of 15.9624, and a burst duration of 11 years. These two categories are the origin categories in the research of phytoremediation of HMs, and they have long been dominant in the research process of phytoremediation of HMs [34,35]. Similarly, on the basis of the subject category co-occurrence network burst detection, “Science & Technology-Other Topics”, which had a burst beginning at the year of 2016, and “Engineering, Multidisciplinary”, “Engineering, Chemical”, and “Green & Sustainable Science & Technology”, with burst beginning at the years of 2014–2016, represent a new trend in the research and development of phytoremediation of HMs, and these categories can be identified as trends in the subject categories. These subject categories are generally applicable, focused on engineering, and comprehensive, and may provide evidence for the application of engineering practices to phytoremediation of HMs [36,37,38,39].

### 3.3. Keywords Co-Occurring Analysis

Keywords can be seen as the soul of an article [25], reflecting the core content of the article in a concise form. Keyword co-occurrence analysis is useful for tracking the development and evolution of research hotspots, as well as obtaining the research trends of a certain knowledge domain. A total of 9823 valid keywords were found in the 6881 search results. According to the keyword co-occurrence analysis results, the top 10 most frequently occurring keywords and the turning points in the network are listed in Table 4 and Table 5. Furthermore, the nodes with strong bursts are listed in Table 6.

Among the top 10 keywords (Table 4), “phytoremediation” was ranked first, with a frequency of 2833, followed by “heavy metals” (2648) and “phytoextraction” (2197). It can be found that the keyword co-occurrence results are greatly influenced by the human factor of the retrieval strategy. For example, “phytoremediation”, “heavy metals”, “phytoextraction”, and “cadmium” were the top four most frequently occurring keywords in the network, and they were also the words in the retrieval strategy. Thus, in this section, the word cloud maps (Figure 6) of the authors’ keywords, keywords plus, titles, and abstracts were drawn on the basis of the text mining function of the Bibliometrix software. In this step, the search terms that appeared in the retrieval strategy were removed from the text mining results, and close synonyms such as “Cu” and “coper”, and singular and plural forms such as “plant” and “plants”, were combined. The high-frequency keywords obtained by the co-occurrence analysis function of CiteSpace greatly interfered with the retrieval strategy. Thus, in this research, the frequency analysis was based on the word cloud result. The most frequent keywords in Figure 6a–d include “accumulation”, “plants”, “contaminated soils”, “tolerance”, “contaminated”, “soils”, “uptake”, “hyperaccumulator”, “*Thlaspi caerulescens*”, “Indian mustard”, “potential”, “hyperaccumulator *Thlaspi caerulescens*”, “toxicity”, “growth”, “effects”, “hyperaccumulation”, “EDTA” (ethylenediaminetetraacetic acid), “rhizosphere”, “translocation factor”, and “bioavailability”. 

According to Table 5, several keywords, such as “*Thlaspi caerulescens*”, “hyperaccumulator”, “phytochelatin”, “organic acid”, and “nickel”, had high BC scores of 0.58, 0.52, 0.52, 0.35, and 0.31, respectively. According to Table 4 and Table 5, keywords such as “soil contamination”, “hyperaccumulator”, “EDTA”, “rhizosphere”, “bioavailability”, “oxidative stress”, “chelating agent”, and “organic acid”, which not only had a high BC score but also had a high frequency of co-occurrence, play key roles in connecting various research topics, have a significant influence on the development of phytoremediation of HMs, and can be recognized as hotspots in the research domain.

Among the 9823 keywords, 72 keywords had occurrence bursts from 1989–2018. Among them, the nodes with high burst strength, long burst duration, and the most recent burst are more attractive to researchers and were listed in Table 6. 

According to Table 6, keywords such as “population” had the highest burst strength (27.1522) from 2001 to 2008. This was followed by “transport” and “plant growth”, with burst strengths of 25.9043 and 25.7854, respectively. Moreover, keywords such as “transport” had the longest burst duration of 16 years, followed by “Brassicaceae” and “cadmium uptake”, with a burst duration of 11 and 10 years, respectively. These findings indicate the prominent role of these research subjects in the phytoremediation of HMs. It was worth mention that several keywords occurred burstness in recent years: “plant growth” and “water” burst in 2015, with burst strengths of 25.7854 and 19.2366, respectively, and a burst duration of 4 years. Keywords such as “bioma”, “bacteria”, and “mine tailing” had a burst period from 2016 to 2018 and burst strengths of 18.7335, 12.9124, and 8.3637, respectively. This represents the attention paid by peer researchers to these fields and the research trends in recent years.

Furthermore, the characteristics of the knowledge structure evolution over time in the field of phytoremediation of HMs were explored on the basis of the keywords’ burst times.

In the period from 1989 to 1999, there was only one keyword (“plant”) that had an occurrence burst. However, there were 38 keywords with occurrence bursts from 2000 to 2010. In this period, keywords with occurrence bursts can be divided into three groups:The plants or hyperaccumulator group included the keywords *Thlaspi caerulescens*, *Arabidopsis*, *Silene vulgaris*, *Pteris vittata* L, *Salix*, *Zea may* L, nickel hyperaccumulator, Brassicaceae, *Holcus lanatus* L, and Indian mustard.The remediation mechanisms group (including physiology and biochemistry of plants) included the keywords metal tolerance, localization, compartmentation, leaves, HM detoxification, glutathione, response, cellular compartmentation, phytochelatin, iron, phytotoxicity, and expression.The enhancements technology group included the keywords *Saccharomyces cerevisiae*, phosphorus, revegetation, population, phosphate, assisted phytoextraction, and chelating agent.

In the period of 2011 to 2018, 32 keywords had occurrence bursts, and these keywords could be divided into four groups:The plants or hyperaccumulator group included the keywords *Helianthus annuus*, hyperaccumulator *Thlaspi caerulescens*, *Arabidopsis halleri*, fern, *Zea mays* L, and *Sedum alfredii*.The enhancement technology group (exogenic substances) included the keywords citric acid, trace element, organic acid, EDD, availability, and enhanced phytoextraction.The enhancement technology group (bacteria/microorganisms) included the keywords bioma, bacteria, *Arbuscular mycorrhiza*, plant growth, and amendment.The engineering/field application group included the keywords phytomining and mine tailing.

Overall, considering the burstness detection results of the keyword co-occurrences, it can be concluded that the research hotspots of phytoremediation of HMs mainly have the following four points: HM hyperaccumulators, uptake mechanisms of HMs, the enhancements of technology for HM uptake, and the engineering/field application of phytoremediation of HMs.

The cluster analysis of the keyword co-occurring network can be used to obtain the distribution of the research on phytoremediation of HMs, and the timeline view can be used to more intuitively analyze its evolution (Figure 7). In this research, the keyword co-occurrence network is divided into 12 co-citation clusters. On the basis of the log-likelihood ratio (LLR) algorithm and according to the titles of their own citing articles, the co-citation clusters were obtained and automatically labeled with the format “# + number + Label” with CiteSpace. As shown in Figure 7, we were more interested in the clusters with colored labels. The mean years of these clusters were distributed over the last decade and they were defined as the active clusters in this study. We were particularly interested in these clusters because they are more likely to represent the research trends of phytoremediation of HMs.

In this research, the co-occurrence network clustering results had a mean *Q* value of 0.7747 and a mean *S* value of 0.9396, which indicated high reliability of the results. The largest cluster (#0) had 19 members, and the *S* value was 0.947. It was labeled as “arsenic hyperaccumulator” by the LLR algorithm, “effect” by the TFIDF (term frequency-inverse document frequency) analysis, and “plant” by the MI (mutual information) analysis. The most active citation to cluster #0 was a research paper on the effects of different single-metal concentrations of Cd, Ni, and Cu and their subcellular distribution in *Brassica juncea* L. var. megarrhiza [40]. The second-largest cluster (#1) had 18 members and an S value of 0.869. It was labeled as “novel mechanism” by LLR, “plant” by TFIDF, and “different metal-bearing solid” by MI. It had two active citations: a research paper on zinc hyperaccumulation and cellular distribution in *Arabidopsis helleri* [41], and a research article that reports a newly observed novel mechanism of silicone uptake in plants (Zhao et al. 2000). This reflected that the mechanism of phytoremediation of HMs is the biggest research hotspot in this topic, and it is consistent with the conclusion of that based on the burst detection of the keywords. The third-largest cluster (#2) had 15 members and an S value of 0.762. It was labeled as “Pb-EDTA accumulation” by LLR, “zinc” by TFIDF, and “*Nicotiana tabacum*” by MI. The most active citation to cluster #2 was research on the effect of EDTA on maize seedlings’ response to cadmium-induced stress [42]. This reflects that the technological enhancements for phytoremediation of HMs is another research hotspot in this topic, and it is also consistent with the conclusion of that based on the burst detection of the keywords.

According to the timeline view of the co-occurrence network clustering results (Figure 7), from 2017 to 2018, a total of four keywords (such as “wastewater” in cluster #0, “induced oxidative stress”, and “serpentine soils” in cluster #1, and “contamination” in cluster #7) had occurrence bursts from 2017 to 2018, which may represent the research trends in this field to some extent.

### 3.4. Reference Co-Citation Analysis

Scientific research needs to be based on the knowledge accumulated from relevant previous research. In other words, subsequently published papers usually rely on previously published literature and research results within a given subject or other related subjects as its references. It is generally believed that two papers that appear simultaneously in the same reference list have a co-citation relationship. Papers with co-citation relationships usually have intrinsic relations, and thus the relationship and structure in the academic field can be revealed with co-citation analysis.

In this research, a total of 6856 of 6881 publications in the local database had global citation records, whereas 1730 publications had local citation records. There was a total of 129,313 citation records, including 4339 references in the local database. On the basis of the analysis of the co-cited references from the documents, Figure 8a displays a 404-node, 494-link reference co-citation network of the research on the topic of phytoremediation of HMs from 1989 to 2018. The reference co-cited network after the burst detection is displayed in Figure 8b. In Figure 8a, the colored lines represent the first co-citation that occurred in the last decade. The gray line represents that of 10 years ago.

There were several documents marked with purple circles or red centers, usually with a large BC score or high burstness, suggesting that they are probably landmark papers in the field and have attracted the most extraordinary degree of attention from its scientific community.

Thus, the nodes in the middle of Figure 8a represent the early studies on the phytoremediation of HMs [43,44,45], as the dominant color is gray (gray nodes and gray links), whereas both sides of the network, which are dominated by other colors (blue, green, yellow, or red links and colored nodes) represent the recent research [46,47,48]. There are a large number of nodes with a large size in the early research period and some marked with purple rings, which represent the rich and extensive knowledge structure in the research of phytoremediation of HMs. The large number of large nodes appearing on both sides of the network represents the rapid development of this field in recent years. Furthermore, on the basis of the co-cited reference analysis, the burst detection, cluster analysis, and timeline view were carried out and displayed the evolution of the overall trends in the research of phytoremediation of HMs (Figure 9).

According to the reference co-citation analysis result, the top 10 most frequently co-cited references (nodes with a large size) and the turning points (nodes with a high BC score) of the reference co-citation network from 1989 to 2018 are displayed in Table 7 and Table 8. Among 129,313 references, 235 references had occurrence bursts from 1989 to 2018. The top 10 strongest citation bursts are shown in Table 9.

The most frequently co-cited document, with a co-citation frequency of 277, was a review article focused on the background, concepts, and future trends in the research of phytoremediation of HMs [16] (listed at the top of Table 7). This was followed by two other review articles about hyperaccumulators. van der Ent et al. clarifies the conditions for the use of the term hyperaccumulator and (re)defines some of the hyperaccumulator terminology [49]; this publication had a co-citation frequency of 260. Krämer outlines the hyperaccumulator germplasm and reviewed the physiological, molecular, and genetic basis underlying metal hyperaccumulation and its evolution [50]. These three review articles summarize the shortcomings of the current research, analyze the future trends of phytoremediation of HMs, and standardize the concepts and terminology related to phytoremediation and hyperaccumulators and the identification of a hyperaccumulator. These two pieces of research regulate the use of terms related to phytoremediation and hyperaccumulators and greatly promote research related to the phytoremediation of HMs.

The top-ranked document by BC score, with a BC score of 0.65, was a research article that was published in *Nature* in 2008 (Table 8) and clarified the contribution of HMA4 (heavy metal ATPase 4) to metal hyperaccumulation or hypertolerance [57]. Van Zaal contributed the publication with the second-highest BC score (0.58) in 1999; he reported a novel *ZAT* (Zn transporter of *Arabidopsis thaliana*) gene that is involved in the transport of Zn in *Arabidopsis* [58]. The publication with the third-highest BC score (0.55) was contributed by Wu in 1999 and focused on the effect of exogenous chelate on the availability, uptake, and translocation of lead in the phytoextraction of Pb [59].

An article by Ali et al. in 2013 had the strongest citation burst [16]. Its burst lasted for 5 years, from 2014 to 2018, with a burst strength of 110.8981 (Table 9). This article was also the most frequently co-cited document and is listed at the top of Table 7. This was followed by the publication by van der Ent et al., which also appeared to be a highly co-cited document, as seen in Table 7. 

In the reference co-citation network, the cluster analysis function divided the co-cited references into several clusters according to the degree of the close association between the references. The references that occurred within the same clusters were tightly connected to each other and loosely connected to the other clusters. In this research, the reference co-citation network was divided into 19 co-citation clusters. The cluster results with burst detection are shown in Figure 9a. The clustering results had a mean *Q* value of 0.8828 and a mean *S* value of 0.9354, which indicates a high reliability of the results. On the basis of the LLR algorithm and according to the titles of their own citing articles, the reference co-citation clusters were obtained and automatically labeled with CiteSpace with the format “# + number + Label”. As shown in Figure 9a, we found that in the past 10 years, a total of nine co-citation clusters have occurred. They were arranged according to the cluster size in Figure 9b,c. These clusters were defined as the active clusters in this study. We were particularly interested in these clusters because they were more likely to represent the research trends of the phytoremediation of HMs, and these clusters were analyzed in detail as follows:

The largest cluster (#0) had 40 members, and the *S* value was 0.917. It was labeled as “EDDs deficiency” by LLR, had a mean year of 2001, and the most active citation in this cluster was a review article on the application of field crops in the phytoremediation of metal-contaminated land [68]. In this review paper, the advantages and disadvantages of crops as HM pollution remediation plants were reviewed by Vamerali et al., and some technical measures that may be beneficial to improving the enrichment ability of crops for HMs were proposed. Tao investigated the effects of ethylenediaminetetraacetic acid (EDTA), triethanolamine (TEA), and citric acid on the extractability of metals from soil and HM uptake and accumulation in *Glycine max* L. [69]. Zaier reported that the exogenous addition of EDTA can significantly enhance shoot HM accumulation in *Brassica napus* [70]. Cluster #0 has the most members and contains a large number of large nodes and a series of nodes with strong citation bursts; thus, cluster #0 is one of the research hotspots in the field of phytoremediation of HMs. This cluster mainly focused on the effects of exogenous chelating agents on the enrichment of HMs in plants.

As the largest active cluster in references co-cited network, cluster (#1) was labeled as “*Arabidopsis thaliana*” by LLR, with the mean year of 2006 and an *S* value of 0.958. It had 27 members and the most active citer to the cluster was a review article on HMs hyperaccumulator [46]. Rascio and Navari-Izzo reviewed the process and mechanism of plant hyperaccumulation of HMs. The importance of the regulation and expression of genes (such as members of the *ZIP* (*Zinc-regulated transporter Iron-regulated transporter Proteins*)*, HMA* (*Heavy Metal transporting ATPases*)*, MATE* (*Multidrug And Toxin Efflu*)*, YSL*(*Yellow Strip1-Like*)*, and MTP* (*Metal Transporter Proteins*) families) was summarized through comparative physiological and molecular analyses. In this review article by Rascio and Navari-Izzo, two classic enrichment hypotheses of HM uptake, the “elemental defense” and “joint effects” were summarized in detail. The authors also noted that although these hypotheses have been partially confirmed in laboratory experiments, they still need to be further studied to be verified in more plant varieties and more natural environments. Ó Lochlainn’s laboratory research found that tandem duplication and deregulation of HMA4 expression occurred during the process of HM hyperaccumulation in *Arabidopsis halleri* and *Noccaea caerulescens*. This study demonstrated that the parallel evolutionary pathways may be the basis for these two occurrences of Zn/Cd hyperaccumulation in the Brassicaceae and pointed out that novel cis-regulatory elements help to increase the expression of the HMA4 gene in *N. caerulescens* [71]. Cluster #1 had relatively more members, a later average year, and more nodes with strong citation bursts. It is reasonable to assume that the research field of cluster #1, which represents the phytoremediation mechanism/process of HMs in hyperaccumulators, represents one of the research trends of phytoremediation of HMs.

Cluster #2 was the second-largest active cluster in the reference co-cited network, with a mean year of 2008, an *S* value of 0.983, and 24 members. It was labeled as “plant-associated bacteria” by LLR. The most active citation in the cluster was a research article on the promoting effect of endophytes on the uptake of HMs [72]. The second most active citation in the cluster was a review article about the bacterial and fungal microbiota of hyperaccumulator plants [73]. Thijs et al. provided insights into the research status of the potential for the typical plant-associated microbiota of hyperaccumulator plants for enhancing metal uptake by plants and discussed the positive impact of microbial-enhanced metal phytoremediation and phytomining. Therefore, cluster #2 represents the trend of research on the plant-associated microbiota-enhanced HM uptake for phytoremediation of HMs.

Cluster #3 was the newest active cluster in the reference co-cited network, with a mean year of 2012, an *S* value of 0.977, and 24 members. It was labeled as “ultramafic soil” by LLR. The most active citation in the cluster was a research article on the agromining potential of nickel-hyperaccumulating plants [74]. The field experiment of Pardo et al. found that nickel hyperaccumulators such as *Alyssum murale* and *Leptoplax emarginata* both have good application potential for nickel agromining. The second most active citation in cluster #3 was a review article on agromining systems for nickel recovery [75]. Kidd reviewed cases of the implementation of agromining engineering practices for the restoration of natural metalliferous soils and summarized the positive effects of fertilization regimes, crop selection and cropping patterns, and bioaugmentation in nickel agromining. At the same time, Kidd pointed out that the subsequent development of this technology needs to be improved in the posttreatment of the hyperaccumulator biomass, and the hydrometallurgical process should be considered to replace the pyrometallurgical process [76]. Therefore, cluster #3 represents the trend of research on agromining for phytoremediation of HMs.

Cluster #4 was the fourth largest active cluster in the reference co-cited network, with a mean year of 2007, an *S* value of 0.974, and 24 members. It was labeled as “microbial communities” by LLR. The most active citation to the cluster was a research article about the effect of culturable bacteria on plant growth and HM availability [77]. Kuffuner et al. found that the bacteria associated with Zn/Cd-accumulating *Salix caprea* had a reliable effect on promoting the growth of *S. caprea* and the potential to increase the uptake of Zn/Cd. The second active citation in this cluster was a review article about the approaches for enhanced phytoextraction of HMs [47], in which the authors reviewed metal tolerance and accumulation mechanisms in plants and explored the effects of environmental and genetic factors on HM uptake. Thus, cluster #4 represents the trend of research identifying approaches for enhanced phytoextraction of HMs for phytoremediation.

Clusters #6, #7, and #9 were all focused on HM hyperaccumulators. The most active citation in cluster #6 was a part of the book series “Reviews of Environmental Contamination and Toxicology” that focuses on the phytoextraction of Cd [78]. Shahid et al. comprehensively reviewed the biogeochemical behavior of Cd in soil–plant systems. This article was also the most active citation in cluster #9. The second most active citation in cluster #6 was a research article about the enhancing effect of an exogenous substance on HM uptake in remediations of HM-contaminated soil [79]. Dary et al. evaluated the effects of *Bradyrhizobium* sp. 750 and HM-resistant PGPR (plant growth promoting rhizobacteria) on the reclamation of multimetal-contaminated soil by *Lupinus luteus*. The second active citation in cluster #9 was a research article about the tolerance and accumulation characteristics of *Siegesbeckia orientalis L*. on cadmium [80]. The most active citation in cluster #7 was a review article on hyperaccumulators [46]. The second most active citation in cluster #7 was a research article on a cadmium hyperaccumulator that forces the cellular sequestration of Cd in *Sedum alfredii* [81]. Thus, clusters #6, #7, and #9 represent the research trends of phytoremediation of HMs by hyperaccumulators.

Cluster #10 was the 11th largest active cluster in the reference co-cited network, with a mean year of 2012, an *S* value of 0.907, and 20 members. It was labeled as “castor bean” by LLR. The most active citation in the cluster was a review article on the positive role of plant-associated microbes in the process of phytoextraction of HMs [82]; this review highlighted the plant–microbe–metal interaction in the phytoextraction process. Pérez-Palacios constructed a double genetically modified symbiotic system to enhance the phytostabilization of copper in legume roots [83]. Thus, cluster #10 represents the trend of research on the enhancing effect of plant-associated microbes on HM uptake in phytoremediation of HMs pollution.

The 14th largest cluster (#13) had 19 members and an *S* value of 0.974. It was labeled as “*Brassica napus*” by LLR, phytoextraction by TFIDF, and emerging technology by MI. The most active citation in the cluster was a research article on the enhanced effect of chemical amendments (rhamnolipid, citric acid, and EDDs) on the phytoremediation of HMs [84]. Zaier et al. reported another study on the effects of EDTA on the phytoextraction of HMs [70]. Thus, cluster #14 represents the trend of research on the enhancing effect of exogenous substances on HM uptake in phytoremediation.

## 4. Discussion

On the basis of the WoSCC database, this study investigated the research hotspots and developmental trends of research on phytoremediation of HMs by using bibliometric and scientometric analysis and CiteSpace and Bibliometrix software. Because the continuous research on the phytoremediation of HMs in the WoSCC database began in 1989, the research time in this study was limited to 1989–2018. Similarly, some important research results were not included in this research, such as reports on the applied aspects of phytoremediation currently in place (e.g., reports from U.S. EPA (Environmental Protection Agency) reports, government reports on large-scale projects, and COST action (COST action is an intergovernmental framework for European Cooperation in Science and Technology, allowing the coordination of nationally-funded research on a European level.) reports) and the publications in the database (e.g., CNKI (China National Knowledge Infrastructure), Scopus). Despite these limitations, we believe that this research may inform future research and encourage the sustainable development of technology for phytoremediation of HMs. This research method may be helpful for new researchers who need to quickly integrate into a research area and provide a new alternative to exploring research trends on specific topics.

Phytoremediation is considered an eco-friendly, low-cost pollution remediation technology and has received extensive attention for the remediation of HM pollution. The number of publications non-linearly increased over time. This same trend was observed in Koelmel et al. [85]. The linear model can better describe the research trends in phytoremediation of HMs. In the research periods of 1989–1999, 2000–2010, and 2011–2018, the linear models explained 84.1%, 96.42%, and 81.05% of the variation in phytoremediation of HMs, respectively. Using the model of the research periods of 2011–2018 would predict that the research focusing on phytoremediation of HMs would increase from approximately 466 in 2011 to 705 in 2019. As research on the phytoremediation of HMs has obvious interdisciplinary characteristics, in this research, the panoramic network of the subject categories of HM phytoremediation research from 1989–2018 was generated with the co-occurrence analysis function of CiteSpace. The interdisciplinary characteristics and the basic, crucial, and emerging subject categories were obtained by an analysis of the indexes (frequency, BC score, and burstness) of the co-occurrence network. In 2013, Ali et al. summarized the interdisciplinary characteristics of phytoremediation research [16]. The author listed the basic disciplines involved in phytoremediation, including environmental engineering, soil chemistry, ecology, plant biology, and soil microbiology. By analyzing the co-occurrence of the subject categories, we can not only gather more detailed information about the interdisciplinary subject categories, but also detect the intrinsic associations between the subject categories. It is worth mentioning that the interdisciplinary results obtained from the co-occurrence analysis were consistent with the results obtained by senior experts on the basis of their years of research experience.

The co-occurrence of keywords and the burst detection of keywords have been widely used for identifying research hotspots and trends [86,87,88,89,90]. Usually, keywords that have a high frequency, high BC scores, or the strongest burst provide a reasonable description of research hotspots. Moreover, among these keywords, those that occurred in recent years represent the research trends. However, because the local database used for the keyword co-occurrence analysis is constructed on the basis of a search term, the frequency of the search term is abnormally high in the network [86,91]. In this study, the keyword vocabulary was obtained with the Bibliometrix software. The word cloud maps were drawn with a keyword list that excludes search terms, and this may avoid the effects of human interference on the results.

On the basis of the word cloud maps, keyword co-occurance analysis, and reference co-cited analysis, the current research hotspots and trends were obtained in this research. The phytoremediation of HM soil contamination has received more extensive attention than that of water and air, and it has always been one of the HM phytoremediation research hotspots. In recent years, the phytoremediation of HMs in water has also attracted the attention of researchers, which may represent a new trend in this field [92,93,94,95]. 

Understanding the status of HM hyperaccumulators in the field of HM phytoremediation is crucial. Researchers generally focus on the screening of HM hyperaccumulators and the phytoremediation mechanism/process of HM hyperaccumulators [49,96,97,98]. However, the small number and unclear enrichment mechanism of HM hyperaccumulators have always been a bottleneck restricting the application of hyperaccumulators to HM phytoremediation. Thus, the enrichment hypotheses of HM uptake, such as “elemental defense” and “joint effects”, need to be studied in more hyperaccumulator taxa and realistic growth environments in the field and for more HM elements [46,99]. It is worth mentioning that field crops, which may become a research trend, have attracted the attention of researchers because of their large biomass and easy breeding [68,100,101,102].

The effect of HMs on plant growth is also one of the hotspots of current research (as indicated by keywords such as tolerance, uptake, toxicity, growth, potential, species, EDTA, and rhizosphere) [103,104,105]. In particular, there has been a focus on the effect of the addition of exogenous substances on plant growth and HM uptake in plants [106]. From the perspective of ecological and environmental safety, the ecological hazard of exogenous additives should be fully evaluated, and highly degradable, low/nontoxic exogenous substances are preferred [107,108,109,110].

Enhanced HM phytoremediation technology has been a hot topic in this field. These technologies include the use of exogenous substances (e.g., organic additives, inorganic additives, fertilizers, auxin applications, exogenous inoculation of bacteria, and microorganisms) [111,112,113,114]. Recently, more attention has been paid to the effects of bacteria/microorganisms on the uptake of HMs in plants [115,116,117]. In particular, it is possible to gain insights into the enhanced effect of plant-associated microbes, noncultivatable microorganisms in the roots of hyperaccumulators, and genetically engineered microbes/plants [118,119,120]. It is worth noting that the synergistic action of the soil, plants, and microbes and the mechanisms of metal mobilization, transformation, and detoxification need to be further explored. Genetically engineered microbes/plants may have greater remediation potential, but their impact on ecosystems needs to be elucidated before commercialization.

The theoretical research of HM phytoremediation is aimed at engineering applications, and agromining can not only remediate the environment, but also recover valuable metals, which is one of the final research trends of phytoremediation in the future [74,121]. Improving and integrating the upstream and downstream processes of agromining accelerates the application of agromining engineering methods, and the implementation of extension projects indicates the research trends in the field of HM phytoremediation [75,122]. Furthermore, these methods are not only applied to the recovery of HMs as secondary pollutants from industrial activities and the remediation of HMs in contaminated environments, but also to the strategic recovery of metals such as uranium and strontium [123,124,125,126]. 

## 5. Conclusions

Using CiteSpace and Bibliometrix, we summarized the academic output characteristics, interdisciplinarity characteristics, research hotspots, and trends of HM phytoremediation, with a view of the panoramic display of research in this field. This method provides a convenient way for new researchers to systematically and comprehensively review the available research in a field and a method for systematically summarizing the knowledge structure of a subject area and identifying research hotspots and trends. Research on the phytoremediation of HMs has developed rapidly, and it is possible to continue to maintain stable/high-speed development. Research on the phytoremediation of HMs has typical interdisciplinary characteristics. Categories such as “Environmental Sciences & Ecology”, “Plant Sciences”, and “Agriculture” are basic supporting categories of the research on the HM phytoremediation. In recent years, scholars of HM phytoremediation have rapidly increased their focus on applied categories such as “Science & Technology—Other Topics”, “Engineering, Multidisciplinary”, “Engineering, Chemical”, and “Green & Sustainable Science & Technology”, which may mean the approach of the application of engineering to HM phytoremediation. At present, research hotspots in the field of phytoremediation of HMs mainly focus on the remediation of soil pollution. “Hyperaccumulator”, “enrichment mechanism/process”, “enhanced technology”, “synergistic soil-plant-microbe action”, and “the mechanisms of metal mobilization, transformation, and detoxification” are the research hotspots in the field of phytoremediation of HMs. The research trends mainly include phytoremediation of HMs by field crops, enhanced technology for phytoremediation of HMs, improved enrichment mechanisms of HM hyperaccumulators, new and efficient phytoremediation enhancement technologies (e.g., genetically engineered microbes/plants), and engineering applications of phytoremediation of HMs (e.g., agromining).

Due to the limitations of the CiteSpace software on the literature format, this study does not contain all the research results of phytoremediation of HMs. There are some drawbacks to the CiteSpace analysis. For example, different signature formats of the same author cannot be intelligently merged, synonyms cannot be intelligently merged, and the human influence caused by the search method cannot be automatically corrected. In recent years, software writers have gradually improved these deficiencies, and appropriate improvements can be used to reduce the impact of these limitations. In future research, we will further revise these deficiencies, strengthen the analysis, and obtain more detailed and accurate research conclusions.

## Figures and Tables

**Figure 1 ijerph-16-04755-f001:**
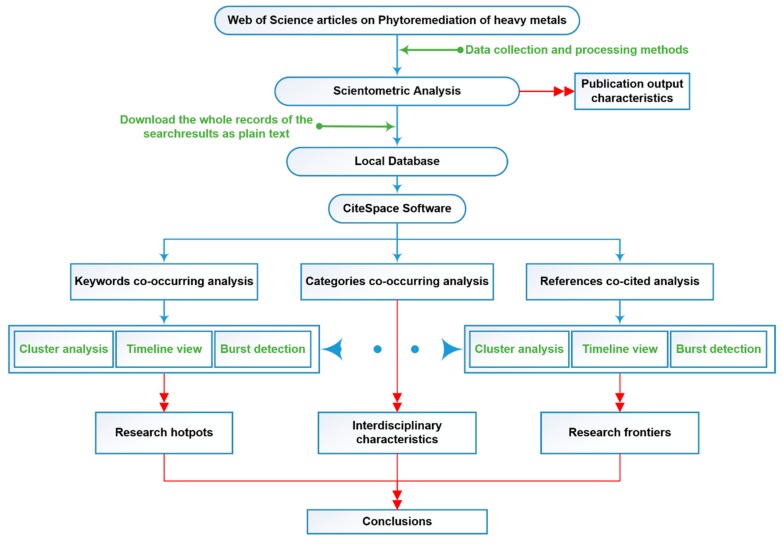
Outline of research design.

**Figure 2 ijerph-16-04755-f002:**
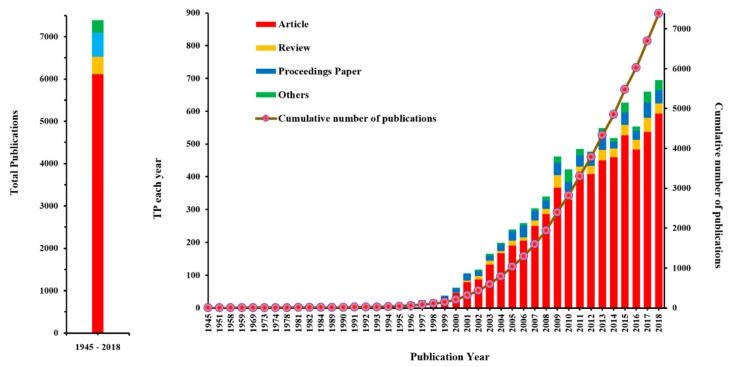
Publication output performance from 1945–2018.

**Figure 3 ijerph-16-04755-f003:**
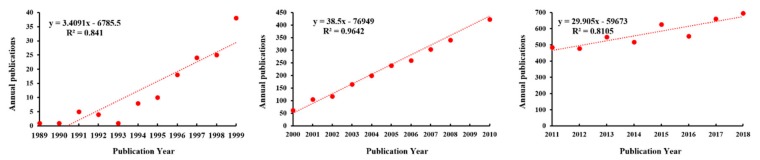
The curve fitting results of annual publications.

**Figure 4 ijerph-16-04755-f004:**
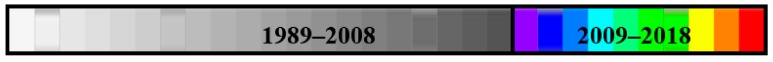
Node/link colors corresponding to the years 1989–2018.

**Figure 5 ijerph-16-04755-f005:**
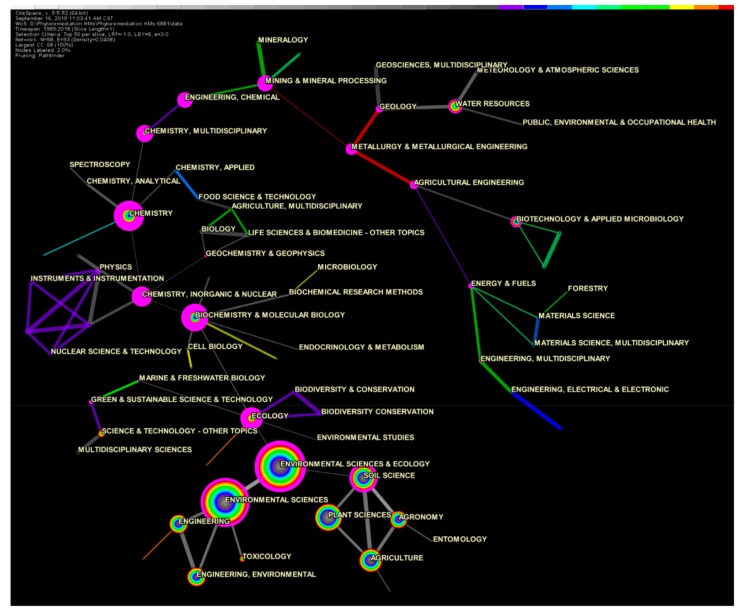
A 68-node 93-link network of subject category co-occurrence from 1989–2018. Note: Each node represents a WoS subject category; the links between the nodes represent the co-occurrence of these two categories in the same publication. Node size represents the co-occurring frequency of this category, and the thicknesses of the line and the tree-ring indicate the levels of the co-occurrence frequency and the numbers of publications, respectively, in a given year.

**Figure 6 ijerph-16-04755-f006:**
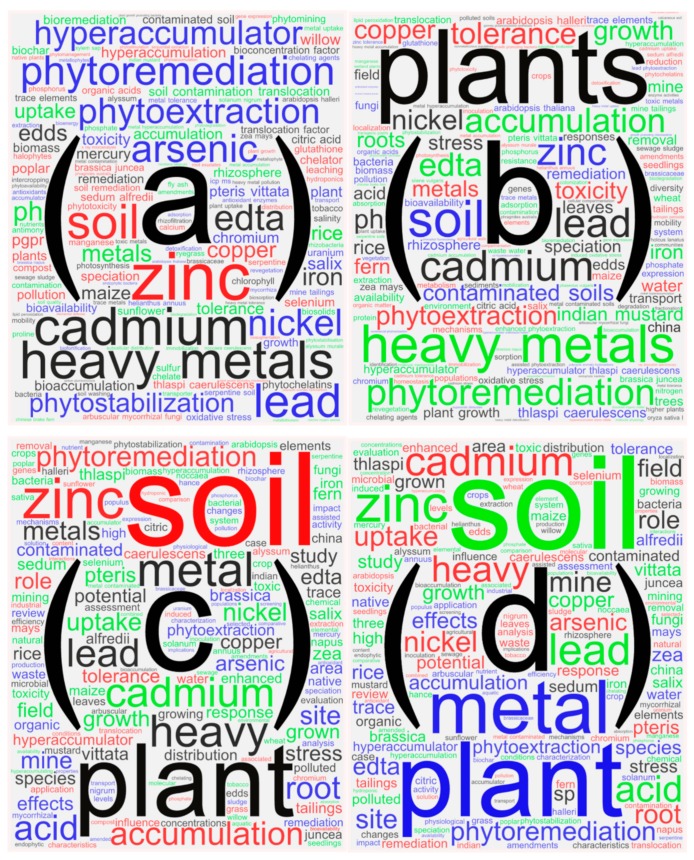
Word cloud maps of most-frequently appeared author’s keywords (**a**), keywords plus (**b**), titles (**c**), and abstracts (**d**) from 1989–2018.

**Figure 7 ijerph-16-04755-f007:**
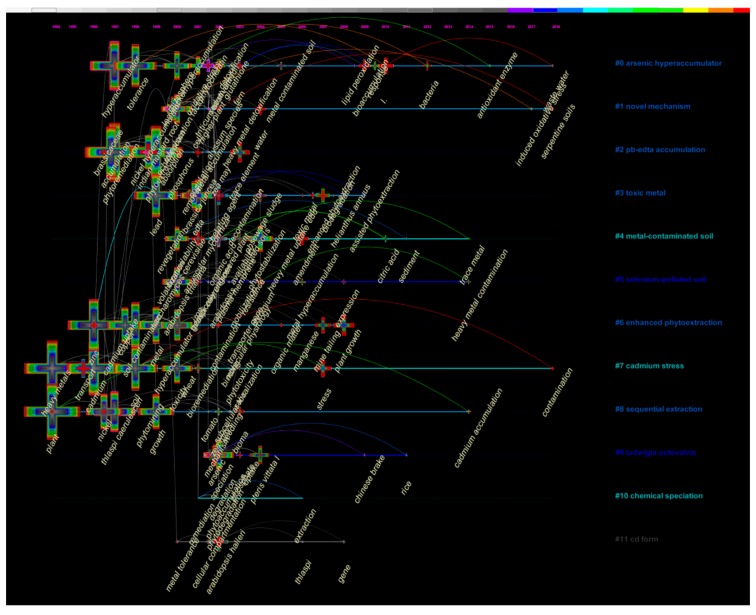
A timeline visualization of the 12 clusters of the keyword co-occurrence network.

**Figure 8 ijerph-16-04755-f008:**
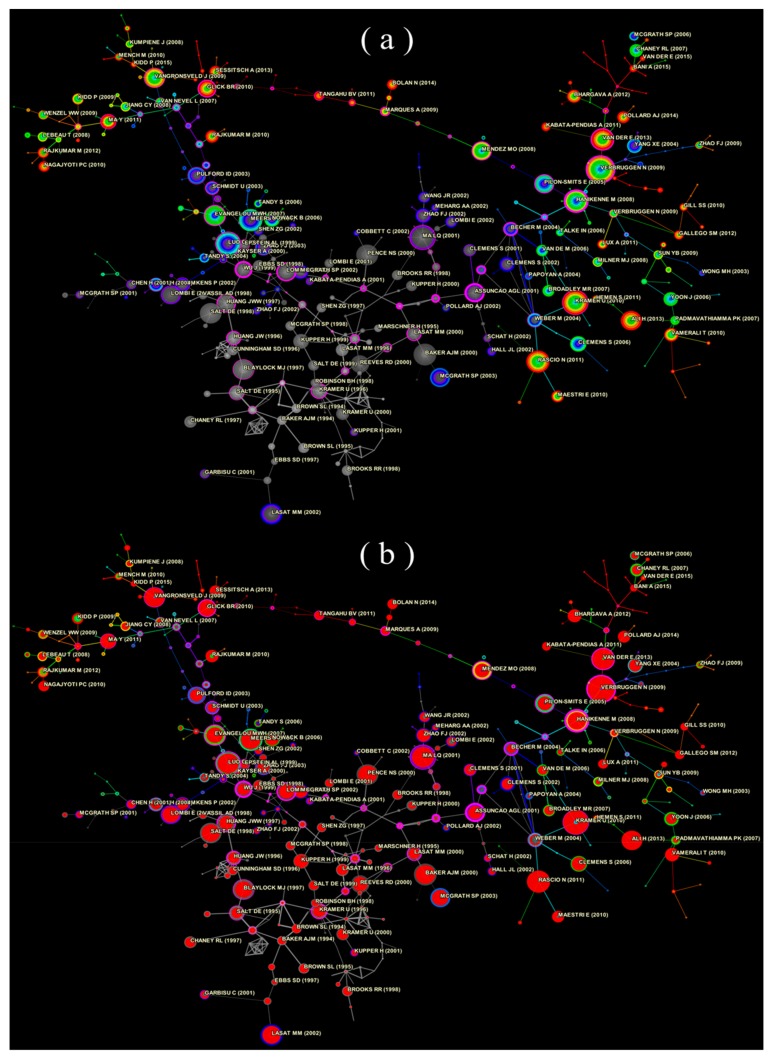
A 404-node, 494-link network of reference co-citation from 1989 to 2018. (**a**): the reference co-citation network. (**b**): the reference co-citation network with burst detection. Note: Each node represents a reference of the publication in the local database, the size of a node is proportional to the number of citation times to the reference. The line between two nodes represents the co-citation relationship between these two references, and the color of the line represents their first together cited in the reference list of phytoremediation of the HM paper. The thickness of the line is proportional to the frequency of their co-occurrence in the reference lists of the paper on phytoremediation of HMs. The node with strong burstness was marked with a red center in (**b**).

**Figure 9 ijerph-16-04755-f009:**
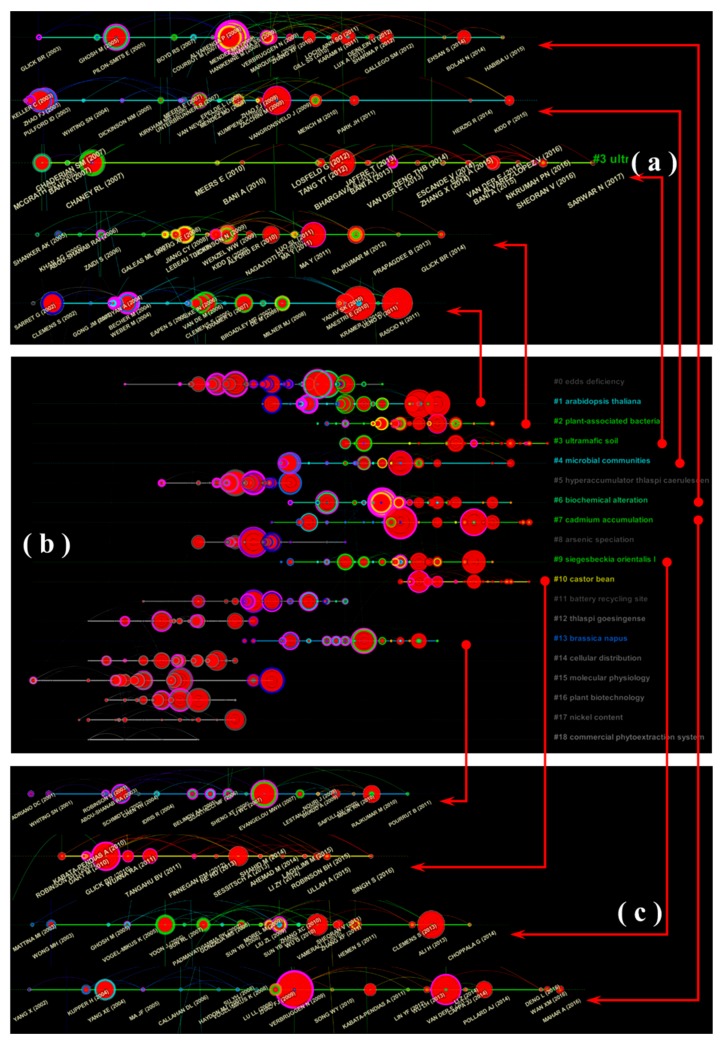
A timeline visualization of the 17 cluster reference co-cited network (**b**), and clusters that represent the research frontier (**a**,**c**).

**Table 1 ijerph-16-04755-t001:** The top 10 subject categories in the co-occurrence network. BC: betweenness centrality.

Rank	Frequency	BC	Mean Year	WoS Categories
1	3725	0.58	1992	Environmental Sciences & Ecology
2	3590	0.23	1995	Environmental Sciences
3	1518	0	1991	Plant Sciences
4	1027	0.06	1997	Agriculture
5	775	0.06	1998	Engineering
6	699	0.28	1997	Soil Science
7	641	0	1998	Engineering, Environmental
8	474	0.06	1997	Agronomy
9	385	0.96	1997	Chemistry
10	337	0.12	2001	Biotechnology & Applied Microbiology

**Table 2 ijerph-16-04755-t002:** Turning points of the subject category co-occurrence network.

Rank	Frequency	BC	Mean Year	WoS Category
1	14	0.98	2004	Chemistry, Inorganic & Nuclear
2	328	0.96	1998	Biochemistry & Molecular Biology
3	385	0.96	1997	Chemistry
4	280	0.96	1999	Ecology
5	144	0.87	1997	Chemistry, Multidisciplinary
6	36	0.86	2013	Mining & Mineral Processing
7	111	0.85	2009	Engineering, Chemical
8	25	0.79	2009	Metallurgy & Metallurgical Engineering
9	3725	0.58	1992	Environmental Sciences & Ecology
10	92	0.53	2006	Agricultural Engineering
11	103	0.34	2003	Energy & Fuels
12	699	0.28	1997	Soil Science
13	99	0.23	2001	Geology
14	3590	0.23	1995	Environmental Sciences
15	57	0.17	2007	Marine & Freshwater Biology
16	86	0.14	2005	Geochemistry & Geophysics
17	19	0.14	2003	Physics
18	20	0.12	2015	Engineering, Multidisciplinary
19	84	0.12	2009	Green & Sustainable Science & Technology
20	337	0.12	2001	Biotechnology & Applied Microbiology
21	313	0.12	2001	Water Resources

**Table 3 ijerph-16-04755-t003:** Among 130 subject categories, 25 subject categories had occurrence bursts from 1989–2018, and the top five of each group were listed.

Ranking Rules	Subject Categories	Strength	Begin	End	Duration
**Ranked by burst strength**	Plant Sciences	57.7754	1991	2006	16
	Agronomy	15.9624	1997	2007	11
	Biochemistry & Molecular Biology	12.8198	2003	2006	4
	Green & Sustainable Science & Technology	11.0896	2014	2018	5
	Multidisciplinary Sciences	9.0432	2001	2002	2
**Ranked by burst duration**	Plant Sciences	57.7754	1991	2006	16
	Agronomy	15.9624	1997	2007	11
	Endocrinology & Metabolism	4.2381	2005	2011	7
	Green & Sustainable Science & Technology	11.0896	2014	2018	5
	Instruments & Instrumentation	4.2559	2009	2013	5
**Ranked by end year of the burst**	Green & Sustainable Science & Technology	11.0896	2014	2018	5
	Engineering, Multidisciplinary	8.8528	2015	2018	4
	Engineering, Chemical	5.4235	2015	2018	4
	Science & Technology—Other Topics	9.0036	2016	2018	3
	Geosciences, Multidisciplinary	4.2049	2015	2016	2
**Ranked by beginning year of the burst**	Science & Technology—Other Topics	9.0036	2016	2018	3
	Engineering, Multidisciplinary	8.8528	2015	2018	4
	Engineering, Chemical	5.4235	2015	2018	4
	Geosciences, Multidisciplinary	4.2049	2015	2016	2
	Green & Sustainable Science & Technology	11.0896	2014	2018	5

**Table 4 ijerph-16-04755-t004:** The top 10 keywords in the co-occurrence network.

Rank	Frequency	BC	Mean Year	Keywords
1	2833	0.03	1997	phytoremediation
2	2648	0.09	1994	heavy metals
3	2197	0.1	1999	phytoextraction
4	2017	0.23	1996	cadmium
5	1998	0.03	1994	plant
6	1876	0.24	1997	accumulation
7	1418	0.27	1996	zinc
8	1378	0.2	1998	soil contamination
9	1280	0.52	1997	hyperaccumulator
10	1105	0.1	1998	soil

**Table 5 ijerph-16-04755-t005:** Turning points of the keyword co-occurrence network.

Rank	Frequency	BC	Mean Year	Keywords
1	730	0.58	1997	*Thlaspi caerulescens*
2	1280	0.52	1997	hyperaccumulator
3	84	0.52	2002	phytochelatin
4	162	0.35	2002	organic acid
5	476	0.31	1997	nickel
6	4	0.3	2002	higher plant
7	3	0.3	2002	*Triticum aestivum* l
8	21	0.28	2000	*Silene vulgaris*
9	1418	0.27	1996	zinc
10	461	0.27	2001	EDTA
11	6	0.27	2000	localization
12	3	0.27	2002	aluminum
13	22	0.26	1999	nickel hyperaccumulator
14	18	0.25	2000	compartmentation
15	1876	0.24	1997	accumulation
16	323	0.24	2001	remediation
17	313	0.24	2000	*Arabidopsis thaliana*
18	2017	0.23	1996	cadmium
19	26	0.23	2002	*Holcus lanatus* L
20	3	0.22	2002	selenium
21	1378	0.2	1998	soil contamination
22	422	0.2	2000	hyperaccumulator *Thlaspi caerulescens*
23	3	0.19	2002	reduction
24	3	0.17	2002	metal accumulation
25	257	0.16	2000	bioavailability
26	26	0.16	2000	saccharomyces cerevisiae
27	208	0.14	2002	chelating agent
28	232	0.12	2001	oxidative stress
29	49	0.12	2007	EDD (Ethylene Diamine Dinitrate)
30	998	0.11	1999	lead
31	149	0.11	1996	transport
32	81	0.11	2000	leave
33	2197	0.1	1999	phytoextraction
34	1105	0.1	1998	soil
35	663	0.1	1999	growth
36	357	0.1	2000	rhizosphere
37	9	0.1	2000	metal tolerance
38	3	0.1	2002	heavy metal tolerance

**Table 6 ijerph-16-04755-t006:** Among 9823 keywords, 72 keywords had occurrence bursts from 1989–2018, and the top 10 of each group were listed.

Ranking Rules	Keywords	Year	Strength	Begin	End	Duration
**Ranked by burst strength**	population	1989	27.1522	2001	2008	8
	transport	1989	25.9043	1996	2011	16
	plant growth	1989	25.7854	2015	2018	4
	Brassicaceae	1989	25.0665	1997	2007	11
	chelating agent	1989	23.093	2006	2010	5
	fern	1989	22.506	2003	2011	9
	lead phytoextraction	1989	21.8002	2001	2008	8
	Indian mustard	1989	21.2228	2005	2010	6
	chromium	1989	20.0711	2013	2016	4
	phytochelatin	1989	19.8803	2002	2006	5
**Ranked by burst duration**	transport	1989	25.9043	1996	2011	16
	Brassicaceae	1989	25.0665	1997	2007	11
	cadmium uptake	1989	15.2189	1997	2006	10
	fern	1989	22.506	2003	2011	9
	nickel hyperaccumulator	1989	10.6685	1999	2007	9
	population	1989	27.1522	2001	2008	8
	lead phytoextraction	1989	21.8002	2001	2008	8
	*Arabidopsis halleri*	1989	19.444	2004	2011	8
	enhanced phytoextraction	1989	18.2275	2008	2015	8
	*Sedum alfredii*	1989	17.7498	2008	2015	8
**Ranked by end year of the burst**	amendment	1989	15.8994	2014	2018	5
	arbuscular mycorrhiza	1989	13.9994	2014	2018	5
	plant growth	1989	25.7854	2015	2018	4
	water	1989	19.2366	2015	2018	4
	bioaccumulation	1989	16.2449	2015	2018	4
	stress	1989	16.0974	2015	2018	4
	trace element	1989	12.3286	2015	2018	4
	bioma	1989	18.7335	2016	2018	3
	bacteria	1989	12.9124	2016	2018	3
	mine tailing	1989	8.3637	2016	2018	3
**Ranked by beginning year of the burst (top 12)**	bioma	1989	18.7335	2016	2018	3
	bacteria	1989	12.9124	2016	2018	3
	mine tailing	1989	8.3637	2016	2018	3
	mechanism	1989	6.108	2016	2018	3
	plant growth	1989	25.7854	2015	2018	4
	water	1989	19.2366	2015	2018	4
	bioaccumulation	1989	16.2449	2015	2018	4
	stress	1989	16.0974	2015	2018	4
	trace element	1989	12.3286	2015	2018	4
	metal hyperaccumulation	1989	13.6803	2015	2016	2
	amendment	1989	15.8994	2014	2018	5
	*Arbuscular mycorrhiza*	1989	13.9994	2014	2018	5

**Table 7 ijerph-16-04755-t007:** The top 10 most cited reference in the co-citation network.

Rank	Frequency	BC	Mean Year	Cited Reference
1	277	0.04	2013	[16]
2	260	0.25	2013	[49]
3	260	0.08	2010	[50]
4	225	0.15	2001	[51]
5	212	0.03	2011	[46]
6	209	0.42	2009	[52]
7	192	0.03	1998	[53]
8	171	0	2003	[54]
9	170	0.05	2005	[55]
10	169	0.38	2005	[56]

**Table 8 ijerph-16-04755-t008:** Turning points of the reference co-cited network.

Rank	Count	BC	Mean Year	Cited Reference
1	150	0.65	2008	[57]
2	25	0.58	1999	[58]
3	80	0.55	1999	[59]
4	16	0.55	1996	[60]
5	116	0.53	2001	[61]
6	31	0.5	2004	[62]
7	113	0.44	2000	[63]
8	89	0.44	2004	[64]
9	28	0.43	2004	[65]
10	209	0.42	2009	[52]

**Table 9 ijerph-16-04755-t009:** Among 4339 local references, 235 references had occurrence bursts from 1989 to 2018, and the top 10 references with strong burst are listed.

References	Year	Strength	Begin	End	Duration	Cluster
[16]	2013	110.8981	2014	2018	5	#9
[49]	2013	92.5736	2014	2018	5	#7
[53]	1998	79.3831	2000	2006	7	#16
[66]	1997	70.6277	1998	2005	8	#15
[67]	1997	68.7919	1998	2005	8	#16
[51]	2001	67.3511	2002	2009	8	#8
[50]	2010	63.9165	2013	2018	6	#1
[46]	2011	61.9716	2012	2018	7	#1
[43]	1995	56.6835	1996	2003	8	#15
[52]	2009	53.0294	2011	2018	8	#7

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
