# Peer review of "Phytoremediation of Heavy Metal Pollution: A Bibliometric and Scientometric Analysis from 1989 to 2018"

_ijerph, 2019, doi:10.3390/ijerph16234755_

Round 1
Reviewer 1 Report
I'm not sure about the suitability of the present manuscript for their publication. In my opinion, it seems a paper more appropriate for a bibliometrics journal than an environmental sciences journal. For example, this review should be more interesting if the authors said which extractants are more suitable according to each soil type or for each element. The same for plant species or soil types (mining, etc.)
Section 3.4 with clusters it's interesting. Still, for example, for cluster #3 about ultramafic soils, it should be more interesting to know who phytorremediators plants have the best results or who elements are the most phytoextracted.
Author Response
Respond: Thank you for your comments concerning our manuscript entitled “Phytoremediation of heavy metal pollution: A bibliometric and scientometric analysis from 1989 to 2018”(ID:ijerph-633196). Your comments are all valuable and very helpful for revising and improving our paper, as well as the important guiding significance to our researches. We have studied comments carefully and have made corrections which we hope meet with approval.
Revised portions are marked in red in the manuscript. The main corrections in the paper and the response to the reviewer's comments are as flowing:
The main purpose of this paper is to help new researchers quickly integrate into the research of phytoremediation of HMs, and help them more easily master the frontiers and hotpots in this field. At the same time, this paper provides a method to quickly understand a certain research field, which not only is limited to the topic of phytoremediation of HMs research but also can be extended to research in environmental science and related disciplines.
We chose “ijerph” as the target journal for our submission mainly for two aspects. First, this manuscript applies CiteSpace software to obtain the research hotspots and research frontiers of the topic of phytoremediation of HMs. This method can be used in other topics in Environment Science. On the other hand, we found some bibliometric related literature in the ijerph's published records. As you said, the plant's species that can be used for HMs pollution remediation and the applicable soil types for these plants have positive implications for the engineering application of this technology. Similarly, the development of genetically engineered microbes/plants and agromining technology, etc., are also the research hotspots and frontiers in this topic. Since space is limited, we touch on these contents only very briefly. Of course, we will pay more attention to this valuable informations in the follow-up study and conduct in-depth discussions separately in our future research.
Our sincere thanks to you again.
Reviewer 2 Report
comment:
This study presents a review about phytoremediation of HMs in the international context from the Web of Science Core Collection (WoSCC) (1989–2018). This paper aims to evaluate the knowledge landscape of the phytoremediation of heavy metals (HMs) to explore the research hotspots and trends of this field. Two different processing software applications were used, CiteSpace and Bibliometrix which can help new researchers to review the available research in a certain research field. Here are some comments about the manuscript:
In the “introduction” part, authors should explain other relevant analysis methods and highlight the advantages of bibliometric analysis in this study. While clearly expressing the purpose of the research, the significance of this study shoud be explained. In the “materials and methods” part, it is necessary to explain why select the data used for the analysis were only collected from the following Web of Science Core Collection (WoSCC), why not collected from CNKI or others? In the "Results and Discussion" part, the authors just stated the data, the discussion in the article is not very full. It's not a structured discussion. In the "Conclusions" part, the conclusion is poor in summarizing the whole paper. Conclusions should be revised.
Author Response
Respond: Thank you for your comments concerning our manuscript entitled “Phytoremediation of heavy metal pollution: A bibliometric and scientometric analysis from 1989 to 2018”(ID:ijerph-633196). Your comments are all valuable and very helpful for revising and improving our paper, as well as the important guiding significance to our researches. We have studied comments carefully and have made corrections which we hope meet with approval.
Revised portions are marked in red in the manuscript. The main corrections in the paper and the response to the reviewer's comments are as flowing:
Drawing on your comments, the introduction was rewritten. In view of the special requirements of CiteSpace for data structure and content, The the data used for the analysis were collected from the following Web of Science Core Collection. For example, CNKI is mainly in Chinese, and the analysis results of CiteSpace may hinder the reading of non-Chinese readers.More importantly, in this study, co-citation analysis was used to obtain research hotspots and research frontiers. But the current version of CiteSpace does not support the reference analysis of CNKI very well. Your comments remind us that it is necessary to explain this issue, we have explained in sections of “Materials and Methods” and “Discussion”, using the red revision display in the manuscript. Thanks, in the manuscript, the “Discussion” were revised. For ease of analysis, the “Discussion” are listed separately and attached to “Results”. Drawing on your comments, the “Conclusions” were revised.Our sincere thanks to you again.
Reviewer 3 Report
This is a useful and comprehensive review of the literature concerning the phytoremediation of heavy metals. I found the network analysis figure showing nodes of research and how these research topics connect a particularly interesting piece of data analysis. Similarly, the burst analysis was a good way to show how research has evolved over time, both in terms of the academic disciplines contributing to research in phytoremediation and the changes in importance of the topic for each discipline over time. The section on evolution of occurrence bursts of various keywords/groups of keywords was also interesting. However, the data in figure 6-8 was illegible as text is far too small for the reader to interpret. This renders these effectively useless in this format and so the reader relies solely on the interpretations of the authors. The interesting information from this was in the text when certain clusters were described as I could read what the main topics covered in these clusters were e.g. “EDDs deficiency” or “microbial communities”, and the trends of research, but this needs to be clearer from the figures. A number of other suggestions of minor changes are also listed below. With some editing, this would be a paper worth publishing as it would make a valuable contribution to those working in the field of phytoremediation research.
Line 35: rephrase “development of urbanization and industrialization” e.g “urbanization and industrial development” Line 36: rephrase “support the economy and people’s livelihoods” e.g. “support the local and national economy”? Line 37: where the authors first use “HM” abbreviation, instead write in full and give abbreviation in brackets e.g. “Heavy metal (HM)” Line 42: “cause serious risks to the environment” should be re-written e.g. “result in serious negative impacts on the environment” Line 43: rephrase “economic and efficient remediation technology” e.g. “cost-effective and efficient remediation technology” Line 43: rephrase “HM pollution has been receiving worldwide attention” e.g. Line 44: rephrase “cheap and green” e.g. “cheap and sustainable” Line 45: remove “and sustainable” to avoid repetition Line 49: What subject categories? Give examples of the development of key phrases/terminology as research has evolved Line 50: rephrase “received much attention” to “received considerable attention” Line 51-52: rephrase Line 53: Add “Thus, “ to start of sentence (as a consequence of the stage of phytoremediation research, we need this synthesis data) Line 54: rephrase “current status and stay current with the emerging trends” Line 55-56: rephrase “Bibliometric analysis, which is a quantitative method combining mathematical and statistical analyses,” to “Bibliometric analysis - a quantitative method combining mathematical and statistical analyses” Line 59: change “also effectively guides subsequent research” to “can guide subsequent research” Line 59-60: Unnecessary information (e.g. developer/university) included in a definition that doesn’t clearly describe what the software does. Instead, use a definition similar to that given in line 83-84 which is much clearer. Line 99: There appears to be a symbol omitted where you state “In Eq. (1), represents” Line 100: Is there something missing where you state “, and is the number of”? Line 104: rephrase “in the result of network analysis” Line 105: Could you define burstness earlier in the paper? Line 114: Avoid personal pronouns such as “we” Line 116: You don’t clearly state what “pathfinder network scaling pruning” is Line 132: In my opinion, “and so on” is an informal term – change this Line 144-146: I don’t think that including the equation of the line is useful information, at least not in this format (perhaps gradient would be useful?) Line 155-156: This needs to be removed/changed. This is interpretation so belongs in the discussion, but could be seen as overinterpretation of the data as you have not projected what might happen in future and are making assumptions. Line 156-158: This is interpretation of results and so belongs in the discussion. You also provide no evidence to support these claims so could be seen as overinterpretation. Line 159: Although figure 2 is very useful, I do not like that you have three further figures included within the main one. Split these into two figures – one with the main figure ad another with the three separate time periods. Line 171: Repetition of “subject categories” - can you find a different term when you mention this a second time? Do you mean sub-categories? Line 171: You don’t make it clear what you mean by “edge” when you state it is a “93-edge” network – what does this tell me? Line 178: I think that it is worth noting that soil science is also Line 172-179: This is a very interesting diagram; however, some of the descriptive text highlighting main findings from this comes before any information is given on what the colours mean. This needs to come earlier or, even better, be given within the figure itself (see next point). Line 180: The information regarding the colour could be given as a key in the figure itself as it is difficult to follow the text in order to interpret the diagram. The reader should be able to interpret this fairly well from the diagram and title/note alone. Line 204: change personal pronoun “we” Line 208: change text to “also play a vital role” Line 212: Add a line between the table and text Line 231-237: I don’t think that figure 4 (and the text that goes with this) is necessary. This figure is not legible and, in my opinion, does not add anything particularly useful to this paper. Line 265: change personal pronoun “we”. Line 265: “have reason to believe that” is an odd phrase to use here Line 272: remove personal pronoun “we” Line 279: The information in table 4 is far more useful and far easier to understand than figure 4. I do not think that the cluster number is necessary if you remove figure 4. Line 280: I do not like that table 5 is in two halves – it would be better if all 38 lines are in one column rather than split over two. This should not be split over more than one page in the final draft of this paper. Line 283: You give a considerable amount of descriptive text and then provide four tables/figures together, one after another. It would be preferable if the descriptive text for each figure was given and then each figure was shown in turn so that it is easier to follow which section of text is referring to which table/figure. Line 283: Although I appreciate the aesthetic value of the four word cloud maps in figure 5, d in particular, but also a to some extent, is quite difficult to read due to the shape used. Line 286: rephrase “Several keywords burst” Line 286-322: It is unclear which table/figure the authors are using to get this information. It would be useful if these were referred to in the text as appropriate. Please also refer to the previous point about line 283 as these two comments are linked: a lot of text is given here after the figure/table. Line 293: remove personal pronoun “we” Line 296-298 (and rest of document where appropriate): plant species names should be in italics. This goes for all figures/tables that mention plant species and any text that mentions specific species. Line 318-322: This should not be a bullet point. Line 323-393: In my opinion, there is a major issue with this paper in its current form: figure 6, 7 and 8 are illegible and are therefore almost impossible for the reader to follow. These need to be larger and/or simplified, or removed altogether if the information cannot be displayed in an appropriate manner. Line 410: “(Kramer, 2010)” should be referenced in the same way as all other in-text citations Line 460: remove unnecessary comma after “i.e.” Line 485-489: This is a very long sentence Line 490-496: A reference needs to be given for the Kidd paper Line 539: change personal pronoun “we” Line 548: change to “Phytoremediation was considered an eco-friendly, low-cost pollution remediation technology and has received” Line 549-550: Re-write sentence “The subject categories and publications show a rapid increase” – too vague Line 553: Slightly odd terminology “basic, hot and emerging subject categories” – hot categories? Line 572: change personal pronoun “we” Line 572-573: One sentence paragraph – edit this Line 578-622: This section does not require bullet points and would read better without these. Line 596: change start of sentence to “Enhanced HM” Line 608-612: Not particularly useful information Line 625-660: This conclusion is far too long. I also do not like the use of bullet points in a conclusion. This needs to be edited.
Author Response
Respond: Thank you for your comments concerning our manuscript entitled “Phytoremediation of heavy metal pollution: A bibliometric and scientometric analysis from 1989 to 2018”(ID:ijerph-633196). Your comments are all valuable and very helpful for revising and improving our paper, as well as the important guiding significance to our researches. We have studied comments carefully and have made corrections which we hope meet with approval.
Revised portions are marked in red in the manuscript. The main corrections in the paper and the response to the reviewer's comments are as flowing:
Line 35: rephrase “development of urbanization and industrialization” e.g “urbanization and industrial development”
Respond: This part has been revised in the manuscript according to your suggestion.
Line 36: rephrase “support the economy and people’s livelihoods” e.g. “support the local and national economy”?
Respond: This part has been revised in the manuscript according to your suggestion.
Line 37: where the authors first use “HM” abbreviation, instead write in full and give abbreviation in brackets e.g. “Heavy metal (HM)”
Respond: This part has been revised in the manuscript according to your suggestion.
Line 42: “cause serious risks to the environment” should be re-written e.g. “result in serious negative impacts on the environment”
Respond: This part has been revised in the manuscript according to your suggestion.
Line 43: rephrase “economic and efficient remediation technology” e.g. “cost-effective and efficient remediation technology” Line 43: rephrase “HM pollution has been receiving worldwide attention” e.g. Line 44: rephrase “cheap and green” e.g. “cheap and sustainable” Line 45: remove “and sustainable” to avoid repetition
Respond: Respond: This part has been revised in the manuscript according to your suggestion.
Line 49: What subject categories? Give examples of the development of key phrases/terminology as research has evolved.
Respond: “Category” is one of the node types in CiteSpace, and “subject category” is the idiom term of the “Web of Science category”.
Line 50: rephrase “received much attention” to “received considerable attention”
Respond: This part has been revised in the manuscript according to your suggestion.
Line 51-52: rephrase
Respond: We have rephrased “Although phytoremediation of HM pollution has received much attention [13,14], research hotspots and trends based on a perspective that encompasses the entire spatial and temporal record have rarely been studied systematically.” to “Although phytoremediation of HM pollution has received considerable attention [13,14], the research hotspots and trends have rarely been studied systematically. Moreover, no previous studies have analyzed its research corpus to such depth to include aspects such as key-words or co-citation clusters.”
Line 53: Add “Thus, “ to start of sentence (as a consequence of the stage of phytoremediation research, we need this synthesis data) Line 54: rephrase “current status and stay current with the emerging trends”
Respond: Drawing on your comments, this sentence is rewritten as: “Thus, it is imperative to create a summary of the current status with emerging trends and vital turning points in the field of phytoremediation. As a consequence of the stage of phytoremediation research, we need these synthesis data.”
Line 55-56: rephrase “Bibliometric analysis, which is a quantitative method combining mathematical and statistical analyses,” to “Bibliometric analysis - a quantitative method combining mathematical and statistical analyses”
Respond: This part has been revised in the manuscript according to your suggestion.
Line 59: change “also effectively guides subsequent research” to “can guide subsequent research”
Respond: This part has been revised in the manuscript according to your suggestion.
Line 59-60: Unnecessary information (e.g. developer/university) included in a definition that doesn’t clearly describe what the software does. Instead, use a definition similar to that given in line 83-84 which is much clearer.
Respond: Drawing on your comments, this sentence is rewritten as: “In this research, a scientometric visualization software – CiteSpace was used as a text mining and visualization tool for bibliometric analysis[15].”
Line 99: There appears to be a symbol omitted where you state “In Eq. (1), represents” Line 100: Is there something missing where you state “, and is the number of”?
Respond: Thank you for your correction. Two indicators have been missed in these places and have been added in the manuscript.
Line 104: rephrase “in the result of network analysis”
Respond: This sentence is rewritten as: “Frequency is a counting index in the network analysis results and it reflects the co-occurrence/co-citation frequency of the published articles in the local database [21].”
Line 105: Could you define burstness earlier in the paper?
Respond: We modified the location of the definition of “burstness” in the manuscript and placed it in front of the definition of “Betweenness centrality”.
Line 114: Avoid personal pronouns such as “we”
Respond: This sentence is rewritten as: “In a co-operation/co-citation analysis, the time slice was set as one year, and the top 50 articles from each time slice were selected for analysis.”
Line 116: You don’t clearly state what “pathfinder network scaling pruning” is
Respond: In this paragraph, the definition of “link reduction function” is added.
Line 132: In my opinion, “and so on” is an informal term – change this
Respond: Rephrase “and so on” to “etc.”
Line 144-146: I don’t think that including the equation of the line is useful information, at least not in this format (perhaps gradient would be useful?)
Respond: The equations 2-4 were deleted, as it repeats with Figure 3 (New figure that obtained from the original Figure 2).
Line 155-156: This needs to be removed/changed. This is interpretation so belongs in the discussion, but could be seen as overinterpretation of the data as you have not projected what might happen in future and are making assumptions. Line 156-158: This is interpretation of results and so belongs in the discussion. You also provide no evidence to support these claims so could be seen as overinterpretation.
Respond: These sentences were deleted.
Line 159: Although figure 2 is very useful, I do not like that you have three further figures included within the main one. Split these into two figures – one with the main figure ad another with the three separate time periods.
Respond: Figure 2 was split into Figure 2 and Figure 3.
Line 171: Repetition of “subject categories” - can you find a different term when you mention this a second time? Do you mean sub-categories?
Respond: “Category” is one of the node types in CiteSpace, and “subject category” is the idiom term of the “Web of Science category”.
Line 171: You don’t make it clear what you mean by “edge” when you state it is a “93-edge” network – what does this tell me?
Respond: In the map of the network, edge refers to the connection between the nodes, in order to avoid misreading, it is corrected as “link”.
Line 178: I think that it is worth noting that soil science is also
Respond: “soil science” also an important subject category of phytoremediation of HMs, and it was classified into the subject category group of “plant science”. Drawing on your comments, we added the main members of the subject categories groups of “Plant Science” and “Environmental Sciences”.
Line 172-179: This is a very interesting diagram; however, some of the descriptive text highlighting main findings from this comes before any information is given on what the colours mean. This needs to come earlier or, even better, be given within the figure itself (see next point). Line 180: The information regarding the colour could be given as a key in the figure itself as it is difficult to follow the text in order to interpret the diagram. The reader should be able to interpret this fairly well from the diagram and title/note alone.
Respond: A description of the color of the node/link is given at the beginning of the paragraph, and Figure 4 is added as a color legend.
Line 204: change personal pronoun “we”
Respond: Drawing on your comments, this sentence is rewritten as: “Thus, according to Table 1 and Table 2, categories such as “Environmental Sciences & Ecology”, “Plant Sciences” and “Agriculture” are basic supporting categories of research on phytoremediation of HMs.”
Line 208: change text to “also play a vital role”
Respond: Drawing on your comments, this sentence is rewritten as: “ “Metallurgy & Metallurgical”, and “Engineering” also play a vital role in the interdisciplinary research of phytoremediation of HMs.”
Line 212: Add a line between the table and text
Respond: Done.
Line 231-237: I don’t think that figure 4 (and the text that goes with this) is necessary.
This figure is not legible and, in my opinion, does not add anything particularly useful to this paper.
Respond: Figure 4 and the text that goes with it were deleted.
Line 265: change personal pronoun “we”.Line 265: “have reason to believe that” is an odd phrase to use here
Respond: “We have reason to believe that” was deleted.
Line 272: remove personal pronoun “we”
Respond: Drawing on your comments, this sentence is rewritten as: “Among the 9823 keywords, 72 keywords have occurrence bursts from 1989-2018. Among them, the nodes with high burst strength, long burst duration, and the most recent burst are more attractive to researchers and were listed in Table 6.”
Line 279: The information in table 4 is far more useful and far easier to understand than figure 4. I do not think that the cluster number is necessary if you remove figure 4.
Respond: The cluster number in table 4 was deleted.
Line 280: I do not like that table 5 is in two halves – it would be better if all 38 lines are in one column rather than split over two. This should not be split over more than one page in the final draft of this paper.
Respond: The format of table 5 was revised.
Line 283: You give a considerable amount of descriptive text and then provide four tables/figures together, one after another. It would be preferable if the descriptive text for each figure was given and then each figure was shown in turn so that it is easier to follow which section of text is referring to which table/figure.
Respond: the position of descriptive text for each figure and table was adjusted.
Line 283: Although I appreciate the aesthetic value of the four word cloud maps in figure 5, d in particular, but also a to some extent, is quite difficult to read due to the shape used.
Respond: Figure 5 was redrawn.
Line 286: rephrase “Several keywords burst”
Respond: Rephrase “Several keywords burst” to “It was worth mention that several keywords have occurred burstness in recent years”
Line 286-322: It is unclear which table/figure the authors are using to get this information. It would be useful if these were referred to in the text as appropriate. Please also refer to the previous point about line 283 as these two comments are linked: a lot of text is given here after the figure/table.
Respond: the position of descriptive text for each figure and table was adjusted.
Line 293: remove personal pronoun “we”
Respond: “We found that” was deleted.
Line 296-298 (and rest of document where appropriate): plant species names should be in italics. This goes for all figures/tables that mention plant species and any text that mentions specific species.
Respond: Plant species names in the text and the tables were corrected in italics. As the figures were generated by CiteSpace, the font of plant species names cannot be changed.
Line 318-322: This should not be a bullet point.
Respond: It has been revised.
Line 323-393: In my opinion, there is a major issue with this paper in its current form: figure 6, 7 and 8 are illegible and are therefore almost impossible for the reader to follow. These need to be larger and/or simplified, or removed altogether if the information cannot be displayed in an appropriate manner.
Respond: Figure 6,7 and 8 were redrawn and formatted.
Line 410: “(Kramer, 2010)” should be referenced in the same way as all other in-text citations
Respond: It has been revised.
Line 460: remove unnecessary comma after “i.e.”
Respond: It has been revised.
Line 485-489: This is a very long sentence
Respond: The sentence was rewritten as “The field experiment of Pardo et al. found that, Ni-hyperaccumulators such as Alyssum murale and Leptoplax emarginata both have good application potential for nickel-agromining”.
Line 490-496: A reference needs to be given for the Kidd paper
Respond: It has been revised.
Line 539: change personal pronoun “we”
Respond: It has been revised. And this sentence was rewritten as “Since the continuous research on the phytoremediation of HMs in the WoSCC database began in 1989, the research time in this study was limited to 1989-2018.”
Line 548: change to “Phytoremediation was considered an eco-friendly, low-cost pollution remediation technology and has received”
Respond: It has been revised.
Line 549-550: Re-write sentence “The subject categories and publications show a rapid increase” – too vague
Respond: The sentence was re-wroten as “The number of publications non-linearly increased over time. This same trend was observed in Koelmel et al [80]. The linear model can better describes the research trends in phytoremediation of HMs. In the research periods of 1989 – 1999, 2000 – 2010 and 2011 – 2018, the linear models explained 84.1%,96.42% and 81.05% of the variation in phytoremediation of HMs, respectively. And using the model of the research periods of 2011 – 2018 would predict that the research focusing on phytoremediation of HMs would increase from approximately 466 in 2011 to 705 in 2019.”.
Line 553: Slightly odd terminology “basic, hot and emerging subject categories” – hot categories?
Respond: It has been revised. As the BC score of the node symbolizes the importance of the node, the word “hot” was changed to “crucial”.
Line 572: change personal pronoun “we”
Respond: The sentence was re-wroten as “Based on the word cloud maps, keyword co-occurance analysis, and references co-cited analysis, the current research hotspots and trends were obtained in this research.”
Line 572-573: One sentence paragraph – edit this
Respond: It has been revised.
Line 578-622: This section does not require bullet points and would read better without these.
Respond: It has been revised.
Line 596: change start of sentence to “Enhanced HM”
Respond: It has been revised.
Line 608-612: Not particularly useful information
Respond: This sentence was deleted.
Line 625-660: This conclusion is far too long. I also do not like the use of bullet points in a conclusion. This needs to be edited.
Respond: The conclusion was rewritten.
Our sincere thanks to you again.
Round 2
Reviewer 1 Report
- Line 68. I suggest justifying why the WoS database was selected and not Scopus or Google Scholar with higher coverage and more updated than WoS.
- Delete [cl1] from Line 484
Author Response
Respond: Thank you for your comments concerning our manuscript entitled “Phytoremediation of heavy metal pollution: A bibliometric and scientometric analysis from 1989 to 2018”(ID:ijerph-633196). Your comments are all valuable and very helpful for revising and improving our paper, as well as the important guiding significance to our researches. We have studied comments carefully and have made corrections which we hope meet with approval.
Revised portions are marked in red in the manuscript. The main corrections in the paper and the response to the reviewer's comments are as flowing:
(1)- Line 68. I suggest justifying why the WoS database was selected and not Scopus or Google Scholar with higher coverage and more updated than WoS.
Answer:
In view of the special requirements of CiteSpace for data structure and content, the data used for the analysis were collected from the Web of Science Core Collection. For example, in this study, co-occurrence analysis of subject categories was used to obtain the knowledge basic of phytoremediation of HMs, but the current version of CiteSpace does not support the subject categories co-occurrence analysis with Scopus records very well. And it is difficult to construct literature records that suitable for CiteSpace analysis with the search results from Google Scholar. Your comments remind us that it is necessary to explain this issue, we have explained in sections of “Materials and Methods” and “Discussion”, using the red revision display in the manuscript.
(2)- Delete [cl1] from Line 484
Answer: Thank you for your careful review. The comment of [cl1] was deleted from line 484.

Reviewer 2 Report
I find that the authors have put considerable effort into addressing the reports of the reviewers. As a result the paper is improved and I have no other problem.
Author Response
Thank you very much for taking the time to review our manuscript and thank you for your recognition of our work. Kind regards.